# Reduced Efficiency Roll-Off in White Phosphorescent Organic Light-Emitting Diodes Based on Double Emission Layers

**DOI:** 10.3390/molecules24010211

**Published:** 2019-01-08

**Authors:** Ren Sheng, Ying Gao, Asu Li, Yu Duan, Yi Zhao, Jie Zheng, Ping Chen

**Affiliations:** 1State Key Laboratory on Integrated Optoelectronics, College of Electronic Science and Engineering, Jilin University, Changchun 130012, China; poiuytr_000@sina.com (R.S.); lias16@mails.jlu.edu.cn (A.L.); duanyu@jlu.edu.cn (Y.D.); zhaoyi2688@163.com (Y.Z.); zhengjie@jlu.edu.cn (J.Z.); 2Institute of Science and Technology for Opto-Electronic Information, Yantai University, Yantai 264005, China; claragaoying@126.com

**Keywords:** organic light-emitting diodes, double emission layers structure, high efficiency, low efficiency roll-off

## Abstract

We demonstrate high-efficiency white phosphorescent organic light-emitting diodes with low efficiency roll-off. The feature of the device concept is employing two phosphorescent emission layers (EMLs) separated by a mixed interlayer. Both the EMLs are doped by two phosphorescent dyes. The resulting white device with the optimized doping concentration shows a maximum efficiency of 31.0 cd/A with extremely low efficiency roll-off of 30.7 cd/A at 1000 cd/m^2^, 27.2 cd/A at 5000 cd/m^2^, and 25.5 cd/A at 10,000 cd/m^2^, respectively, without any outcoupling structures. This is enabled by the balanced charge carrier transport in EMLs, leading to broader exciton recombination zone.

## 1. Introduction

With 30 years of development, organic light-emitting diode (OLED) has been utilized into full-color panel display due to its peculiar features, such as high resolution, wide view angle, light weight, and high efficiency [1,2,3]. Generally speaking, white OLEDs can be achieved by employing three primary colors (red, green, and blue) dyes or double complementary colors (blue and yellow) dyes. Additionally, the three primary colors strategy is much superior because the spectrum covers the whole visible region [4,5]. Multifarious device architectures have been reported to achieve efficient white OLEDs, such as single-emission layer (EML) structure with multiple doped dyes, multiple EMLs structure, stacked or tandem structure [6,7,8]. Nowadays, phosphorescent materials have been an essential component of OLEDs because they can harvest both singlet and triplet excitons, leading to 100% internal quantum efficiency. To reduce the severe excitons quenching, phosphorescent dopants are invariably doped into the host materials [9,10,11]. However, the device efficiency still tends to suffer from more precipitous decline with the increase of luminance, which is called efficiency roll-off and impedes white OLEDs in commercial applications [12,13]. Research shows that efficiency roll-off results from a diverse range of effects, such as triplet-triplet annihilation (TTA), singlet-singlet annihilation, singlet-triplet annihilation, triplet-polaron annihilation, and field-induced quenching [14]. Among these effects, TTA is the main influence factor and in urgent need to be solved.

To alleviate the efficiency roll-off, enormous efforts have been made. A popular tactic is recombining blue fluorescence with red/orange phosphorescence to fabricate hybrid white OLEDs. For example, Zhang et al. reported highly efficient simplified single-emitting-layer hybrid white OLEDs and achieved an external quantum efficiency roll-off of 5.8% at the luminance of 1000 cd/m^2^ [15]. With a mixed host, or double emission layers structure, OLEDs can efficiently extend their exciton recombination zone and decrease the TTA, leading to reduced efficiency roll-off at high luminance levels. N. Chopra et al. demonstrated high efficiency blue phosphorescent OLEDs based on mixed host architecture with a power efficiency roll-off of 16.7% at the luminance of 1000 cd/m^2^ [16]. G. He et al. demonstrated high-efficiency OLEDs based on a double-emission layer with a power efficiency roll-off of 21.9% at the luminance of 1000 cd/m^2^ [17]. However, the efficiency roll-off of white OLEDs based on all phosphorescent dyes is still far from the requirement of practical lighting applications.

In this paper, we demonstrated highly efficient all phosphorescent white OLEDs with ultra-low efficiency roll-off based on double EMLs structure. The first EML comprised green and red phosphorescent materials doped in 1,3-Bis(carbazol-9-yl)benzene (MCP). The second EML comprised yellow and blue phosphorescent materials doped in MCP. A mixed interlayer was introduced to separate the two EMLs, which facilitates the transport of charge carriers [18,19]. By further optimizing the concentration of blue phosphorescent dye, which shows excellent electron transporting property, balanced carrier distribution and extended recombination zone were achieved. The energy transfer among the emitters was also discussed, which plays a crucial part in decreasing efficiency roll-off. The resulting white device with a low efficiency roll-off of 17.7% at 10,000 cd/m^2^ exhibits maximum efficiencies of 31.0 cd/A and 13.1% without any outcoupling structures. Our finding paves the way for further enhancement of the performance of the white OLEDs.

## 2. Experimental

Figure 1 shows the detailed energy level diagram and the molecular structure of the proposed materials respectively. The pre-patterned indium tin oxide (ITO) coated glass with a sheet resistance of 20 Ω/sq was used as anode. The ITO substrates were scrubbed with acetone, ethylalcohol, deionized water in sequence, and then, cleaned in a UV ozone chamber for 5 min. Subsequently, the ITO substrates were loaded into an evaporation system for deposition. Molybdenum(VI) Oxide (MoO_3_) was used as the hole-injection layer. MCP with good hole transport property acted as the hole transport layer (HTL). 1,3,5-Tri[(3-pyridyl)-phen-3-yl]benzene (TmPyPb) with good electron transport property served as the electron transport layer (ETL). Bis(2-methyldibenzo[*f*,*h*]quinoxaline) (acetylacetonate)iridium(III) (Ir(MDQ)_2_(acac)), Bis(4-phenylthieno[3,2-*c*]pyridinato-N,C2′) acetylacetonate iridium(III) (PO-01), Tris(2-phenylpyridine)iridium(III) (Ir(ppy)_3_), and FIrPic were used as red, orange, green and blue phosphorescent dyes, respectively. Here, MCP was used as the host material due to the high triplet energy level. Finally, Liq covered by 100 nm Al acted as the cathode. The active emissive area of the devices by the crossover of the ITO and Al was approximately 3 × 3 mm. All the devices were fabricated in a high vacuum (3 × 10^−4^ Pa) thermal evaporation chamber. MCP and TmPyPb were grown at the rate of 0.1–0.2 nm/s, while Ir(MDQ)_2_(acac), PO-01, Ir(ppy)_3_, FIrPic, Liq and MoO_3_ were deposited at the rate of 0.02–0.08 Å/s. In addition, Al was deposited at the rate of 0.5 nm/s. The current-voltage-luminance characteristics and Electroluminescent (EL) spectra of the devices without encapsulation were measured by a programmable Keithley 2400 voltage-current source and PR655 spectro-scan spectrometer.

## 3. Results and Discussion

To obtain efficient white OLEDs with ultra-low efficiency roll-off, several strategies were adopted in our experiment. Firstly, double EMLs structure was introduced to widen the excitons formation region. Secondly, a blend of the hole transport material and the electron transport material was employed as the interlayer to prevent dexter energy transfer between the two EMLs and balance charge carriers. Thirdly, MCP and TmPyPb with high triplet energy level (T_1_) were employed to confine triplet excitons and control the location of the exciton recombination region. Here, based on the ideas mentioned above, we fabricated four phosphorescent white OLEDs with the structures of ITO/MoO_3_ (2 nm)/MCP (50 nm)/MCP: 10% Ir(ppy)_3_: 0.5% Ir(MDQ)_2_(acac) (5 nm)/MCP: TmPyPb (1: 1) (1.5 nm)/MCP: X% FIrPic: 0.5% PO-01(15 nm)/TmPyPb (30 nm)/Liq (1 nm)/Al (100 nm). The value of X varied with 5, 10, 15 and 20, corresponding to devices A, B, C and D, respectively.

Figure 2 shows the current efficiency-luminance characteristics of devices A–D. The current efficiency curve shows an initial increase as the doping concentration of FIrPic rises from 5% to 15%, and then decreases with a further increase of the doping concentration to 20%. Device C with Commission International de L’Eclairage (CIE) coordinates (0.38, 0.54) emits warm white light. Note that device C shows maximum efficiencies of 31.0 cd/A and 13.1%. In contrast to devices A and B, the maximum efficiencies are 26.7 and 30.7 cd/A. The detailed electroluminescent characteristics of devices A–D are summarized in Table 1. Figure 3 shows the external quantum efficiency (EQE)-luminance characteristics of devices A–D. The maximum EQE of the four devices are 12.7%, 14.3%, 13.1%, and 9.4%, respectively. Particularly, the EQE still maintains 11.0%, 13.4%, 12.4% and 8.1% at the luminance of 1000 cd/m^2^. As shown in Figure 1, the interfacial energy barrier for electrons injection from TmPyPb (LUMO: 2.7 ev) to MCP (LUMO: 2.4 ev) is 0.3 eV, which results in accumulation of electrons at the interface and, hence, higher turn-on voltage. As the concentration of FIrPic increases, more electrons are injected into FIrPic (LUMO: 2.9 ev) without an interfacial energy barrier. The current density sharply increases as the doping concentration of FIrPic rises, as shown in Figure 4. However, when the FIrPic concentration further increases to 20%, the holes trapped effect becomes serious, resulting in higher turn-on voltage than device C.

As we all know, holes are more mobile than electrons in OLEDs. Therefore, as the doping level of FIrPic rises from 5% to 15%, the increasing electron current induces the presence of more balance carriers and thus promotes efficiency. However, as FIrPic concentrations increase above 15%, excess electrons result in a reduced carrier balance and thus lower efficiency. Moreover, concentration quenching of FIrPic may be another reason for low efficiency [20,21,22,23]. Actually, the maximum efficiencies of device D are only 22.7 cd/A and 9.4%. Furthermore, it must be noted that all devices show extremely low efficiency roll-off. For device C, the current efficiency still maintains 30.7 cd/A at 1000 cd/m^2^, 27.2 cd/A at 5000 cd/m^2^, and 25.5 cd/A at 10,000 cd/m^2^, corresponding to the roll-off of 1.0%, 12.3%, and 17.7%. 

Figure 4 shows the current density-voltage-luminance characteristics of devices A–D. The current density shows increasing tendency from device A to device C and then reduces obviously as the concentration of the FIrPic increases to 20%. The luminance curve shows the identical trend as current density. The variation trend of current density suggests that FIrPic can function to regulate the charge carriers. As is known to all, FIrPic shows great electron injection and transport property. However, FIrPic also possesses the ability of blocking holes [20]. Thus, we made the following speculation. With the FIrPic concentration increasing from 5% to 15%, electron current increases obviously due to the great injection and transport property of FIrPic, the increased number of electrons are more than the reduced number of holes, leading to increased total current density. As the FIrPic concentration increases above 15%, the hole obstruction effect of FIrPic becomes stronger, the increased number of electrons become less than the reduced number of holes, which results in decreased total current density. This phenomenon can be justified by the change tendency of turn-on voltages (voltage at luminance of 1 cd/m^2^) of devices A–D. Device C shows the lowest turn-on voltage of 3.38 V.

Figure 5a shows the normalized electroluminescence (EL) spectra normalized to the blue emission peak of devices A–D at 7 V. It is clear that all the devices show four emission peaks at 470, 509, 550, and 598 nm, originating from FIrPic, Ir(ppy)_3_, PO-01, and Ir(MDQ)_2_(acac), respectively. It is interesting to note that, with the FIrPic concentration increasing from 5% to 15%, the relative emission intensity of red, orange and green sharply increases, and then decreases as the concentration of FIrPic reaches 20%. The phenomenon has also been confirmed in our previous works [24]. With FIrPic concentration increasing from 5% to 15%, more electrons can inject into FIrPic, and the carrier coefficient of utilization is increased. In the four devices, the doping concentrations of PO-01 and Ir(MDQ)_2_(acac) are low (both fixed at 0.5%), so that the emission sites easily reach saturation, leading to the weaker emission intensity than the green emission intensity. Figure 5b exhibits the normalized EL spectra of device C at different luminance. As we can see, the spectra are extremely stable over a large range of luminance. Device C shows a dinky CIE coordinates variation of (0.022, 0.023) from 1501 to 30,000 cd/m^2^.

The emission mechanism of devices A–D is depicted in Figure 6. For blue-orange EML, electrons prefer to inject into FIrPic easily, rather than TmPyPb, due to its lower lowest unoccupied molecular orbital (LUMO). Thus, excitons can directly generate in FIrPic and PO-01. Considering the lower T_1_ of PO-01 than that of FIrPic, efficient energy transfer from FIrPic to PO-01 could certainly occur. Moreover, due to the great electron transport ability of FIrPic, electrons can easily permeate the blue-orange EML and reach the interlayer. The interlayer composed of MCP and TmPyPb with bipolar transport ability improves the transport balance of holes and electrons in each EML, leading to enhanced device efficiency as well as color stability. Furthermore, balanced charge transport also reduced the exciton accumulation, which further expands the exciton recombination zone to reduce TTA behavior. Then, electrons can inject into green-red EML without any obstacles in view of the bipolar transmission characteristics of the interlayer and the lower LUMO of Ir(MDQ)_2_(acac). Moreover, the red emission dye of Ir(MDQ)_2_(acac) owns the ability to trap electrons [25,26], which makes for the direct generation of excitons in green-red EML. Considering the higher T_1_ of Ir(ppy)_3_ and the low doping level of Ir(MDQ)_2_(acac), inefficient energy transfer from Ir(ppy)_3_ to Ir(MDQ)_2_(acac) is inevitable. It is concluded that the both FIrPic molecules with optimized doping concentration and the mixed interlayer can facilitate the charge transport balance simultaneously, which leads to improved efficiency and suppressive efficiency roll-off of devices. The appropriate doping concentration of dopants makes full emission of four colors to achieve white OLEDs with broad spectra. 

## 4. Conclusions

In summary, efficient white OLEDs based on double EMLs with low efficiency roll-off were demonstrated. The two EMLs separated by a mixed interlayer were both doped with two phosphorescent dyes. We found that the blue phosphorescent dye plays an important role in adjusting charge carrier balance. The resulting white device exhibits a maximum current efficiency of 31.0 cd/A with a low efficiency roll-off of 17.7% at 10000 cd/m^2^. We also discussed the emission mechanism and drew conclusions that the high efficiency and low efficiency roll-off were mainly attributed to the broader recombination zone and stable sequential energy transfer among the dopants.

## Figures and Tables

**Figure 1 molecules-24-00211-f001:**
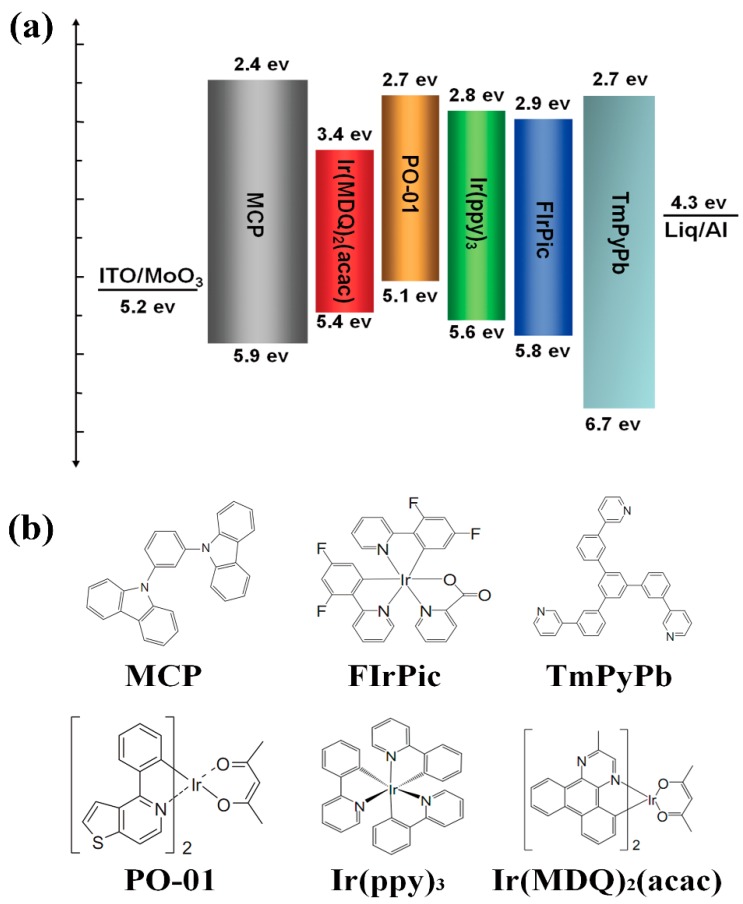
The detailed energy level diagram (**a**) and the molecular structure (**b**) of the proposed materials.

**Figure 2 molecules-24-00211-f002:**
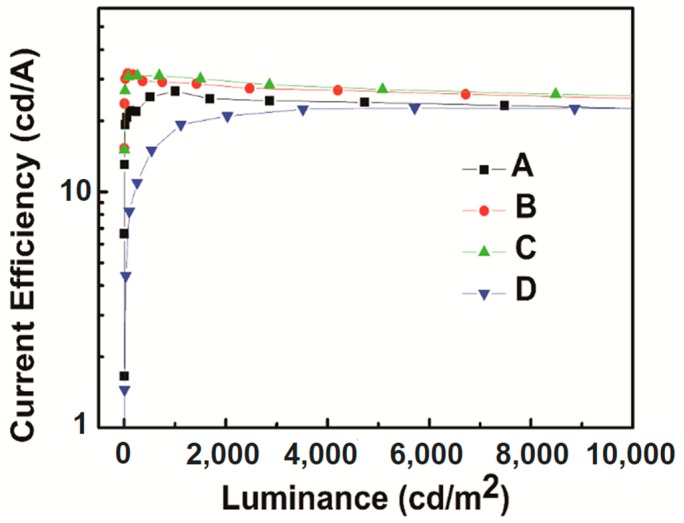
The current efficiency-luminance characteristics of devices A–D.

**Figure 3 molecules-24-00211-f003:**
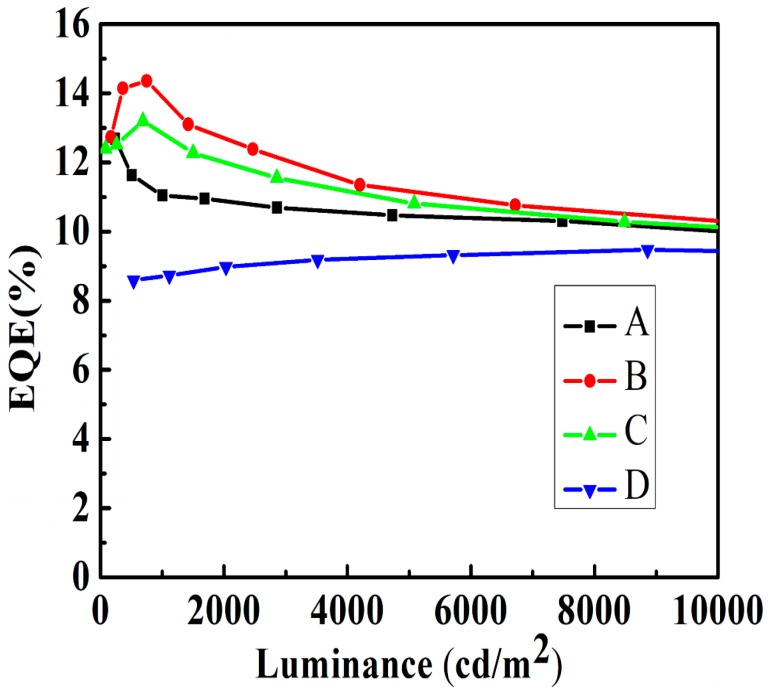
EQE-luminance characteristics.

**Figure 4 molecules-24-00211-f004:**
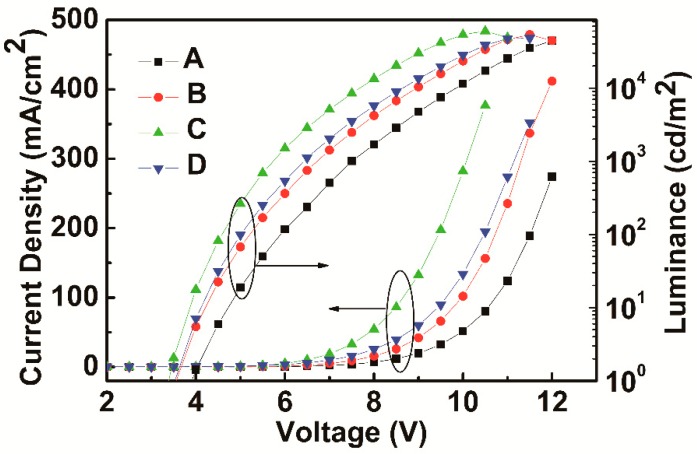
The current density-voltage-luminance characteristics of devices A–D.

**Figure 5 molecules-24-00211-f005:**
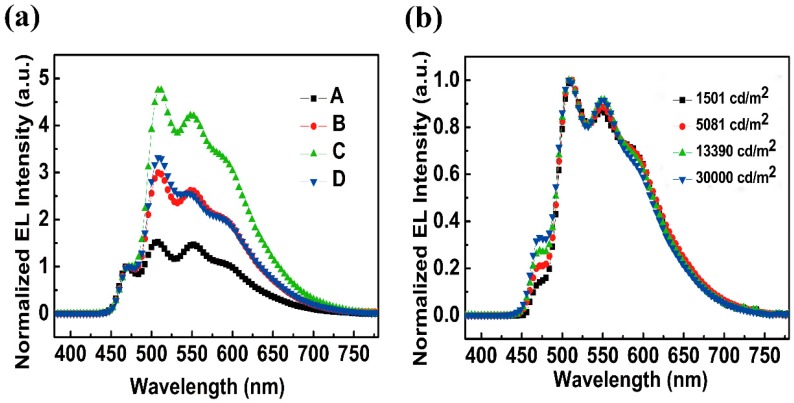
The normalized EL spectra of devices A–D at 7 V (**a**) and the normalized EL spectra of device C at different luminance (**b**).

**Figure 6 molecules-24-00211-f006:**
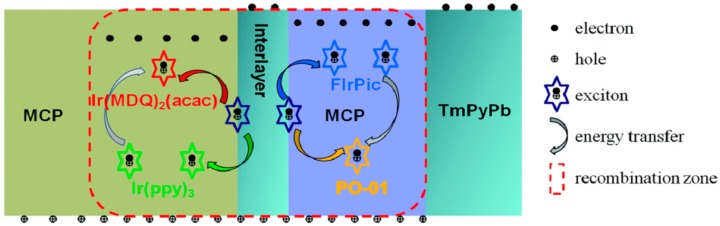
The emission mechanism of white devices A–D.

**Table 1 molecules-24-00211-t001:** Performance characteristics of devices A–D. **η_max_**: Maximum efficiency, **η_1000_**, **η_5000_**, **η_10,000_**: efficiencies at 1000, 5000, and 10,000 cd/m^2^.

Devices	^a^ CIE (x, y)	Turn-on Voltage (V)	Current Efficiency (cd/A)	EQE (%)
			η_max_/η_1000_/η_5000_/η_10,000_	η_max_/η_1000_/η_5000_/η_10,000_
**Device A**	(0.35, 0.48)	3.89	26.7/26.6/23.9/22.6	12.7/11.0/10.4/10.1
**Device B**	(0.36, 0.49)	3.56	30.7/29.0/26.6/24.9	14.3/13.4/11.1/10.3
**Device C**	(0.38, 0.54)	3.38	31.0/30.7/27.2/25.5	13.1/12.4/10.8/10.1
**Device D**	(0.36, 0.52)	3.54	22.7/18.4/22.5/22.5	9.4/ 8.1/9.2/9.2

^a^ CIE coordinates at 1000 cd/m^2^.

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
