# Peer review of "Reduced Efficiency Roll-Off in White Phosphorescent Organic Light-Emitting Diodes Based on Double Emission Layers"

_molecules, 2019, doi:10.3390/molecules24010211_

Round 1

Reviewer 1 Report

The authors report a novel phosphorescent “off-white” OLED with low efficiency roll-off. This result is achieved by a careful design of the device layers and the use of multiple emitters allowing for a large recombination zone. In my opionion, the results shown here are in principle of sufficient novelty and quality for publication in Molecules, but the language has to be thoroughly improved and more data have to be provided before the final decision.

1) The English of the manuscript is sometimes too informal and sometimes incorrect. This makes the content difficult to understand. For example: Page 3, line 98-99: “When the concentration of FIrPic lower than 15%, current density increases with the increase of FirPic concentration.”

There are many more sentences that must be rewritten.

2) According to theCIE coordinates reported by the authors, all the OLEDs are outside the commonly defined white region, as they are more yellowish-green.

3) The color and the low efficiency roll-off efficiency are obtained thanks to multiple emitters with suitable donor/acceptor features. So, in order to make the reader fully appreciate the rationale behind this, the authors should report the photoluminescence emission and absorption spectra of all the Ir-compounds they use as the emitters in the OLED.

4) Total external quantum efficiencies for all the devices should be reported.

p { margin-bottom: 0.25cm; line-height: 115%; }

Author Response

General comment: The authors report a novel phosphorescent “off-white” OLED with low efficiency roll-off. This result is achieved by a careful design of the device layers and the use of multiple emitters allowing for a large recombination zone. In my opionion, the results shown here are in principle of sufficient novelty and quality for publication in Molecules, but the language has to be thoroughly improved and more data have to be provided before the final decision.

Our response: We are grateful to the referee for being favourable in terms of publication of the manuscript in Molecules. The issues indicated by the referees have been addressed, as described in the following responses.

Comment 1: The English of the manuscript is sometimes too informal and sometimes incorrect. This makes the content difficult to understand. For example: Page 3, line 98-99: “When the concentration of FIrPic lower than 15%, current density increases with the increase of FirPic concentration.”There are many more sentences that must be rewritten.

Our response: We appreciate the Reviewer’s suggestion. We have carefully checked the manuscript thoroughly and improved quality of English in the revised manuscript. We hope the Reviewer could be satisfied with the English of revised manuscript.

Comment 2: According to the CIE coordinates reported by the authors, all the OLEDs are outside the commonly defined white region, as they are more yellowish-green.

Our response: Recently, higher efficiency OLEDs with CIE coordinate (x>0.3, y>0.5) were reported, which were defined as warm WOLEDs [J. Mater. Chem. C, 2015, 3, 6359 and Adv. Funct. Mater. 2016, 26, 776.]. It is really true that the CIE coordinates in our work was not satisfied. However, the reported WOLEDs in this work are also identified as warm WOLEDs which have potential use in solid-state light.

Comment 3: The color and the low efficiency roll-off efficiency are obtained thanks to multiple emitters with suitable donor/acceptor features. So, in order to make the reader fully appreciate the rationale behind this, the authors should report the photoluminescence emission and absorption spectra of all the Ir-compounds they use as the emitters in the OLED.”

Our response: Thanks a lot for the thoughtful comments of the Reviewer. We are sorry for not carrying out PL and absorption spectra of all the Ir-compounds measurements because the testing instrument has been out of commission. The used Ir-compounds were bought in Luminescence Technology. The peak value of PL and UV can be checked on the website of http://www.lumtec.com.tw. we also looked through a few reported works which have measured PL and absorption spectra of the Ir-compounds: Opt. Mater. 2008, 31, 366 (PL and absorption spectra of FIrPic), Appl. Phys. Lett. 2002, 80, 2045 (PL and absorption spectra of Ir(ppy)3), J. Phys. D: Appl. Phys. 2015, 48, 365106 (PL and absorption spectra of PO-01), Org. Electron. 2012, 13, 1589 (PL of Ir(MDQ)2(acac)), Isr. J. Chem. 2014, 54, 979 (absorption spectra of Ir(MDQ)2(acac)). Thanks again for the comments, which are really valuable for the improvement of our manuscript.

Comment 4: Total external quantum efficiencies for all the devices should be reported.”

Our response: Thank you very much for this suggestion. The external quantum efficiencies (EQE)-luminance characteristics for devices A-D has been shown in Fig. 3 in the revised manuscript. The EQE for all the devices has also been added in Table 1 in the revised manuscript. Other revisions have also been correspondingly made in the revised manuscript. Special thanks to you for your good comments.

Reviewer 2 Report

This manuscript showed the high-efficiency white phosphorescent organic light-emitting diodes with lower efficiency roll-off. But there still some comments in here.

1.      In the device structure, did the “MCP: TmPyPb (1: 1) (1.5 nm)” as interlayer? Any energy level results of it? Since the interlayer only 1.5nm, whether it is possible the exciton directly transport from EML-1 to EML-2?  

2. In Figure 2, the device C with lowest Voc, why? Low recombination or high electron injection?

3. In Figure 3, compare with device C, I found the Device A and D shown more interesting the results, for example, the roll-off of these devices are much lower than the device C. And the device D got the max current efficiency-luminance at high voltage and luminance, why?  

4. If the 15% is optimized the carrier balance in device, the roll-off also should be lowest in four device, but on the contrary it is highest. Does it mean higher efficiency with lower roll off in this two phosphorescent emission device?

5. Does the Author have another result and experiment data support the electron injection from blue layer to red& green EML, photo-celiv data?

Author Response

General comment: “This manuscript showed the high-efficiency white phosphorescent organic light-emitting diodes with lower efficiency roll-off. But there still some comments in here.

Our response: Thank you very much for your kind work and consideration on publication of our paper.

Comment 1: “In the device structure, did the “MCP: TmPyPb (1: 1) (1.5 nm)” as interlayer? Any energy level results of it? Since the interlayer only 1.5nm, whether it is possible the exciton directly transport from EML-1 to EML-2?

Our response: Thanks for this comment. We used a heterojunction of MCP: TmPyPb (1: 1) as the interlayer located between two EMLs to fabricate the high efficiency and low roll-off WOLEDs. Dexter energy transfer can be prevented by the thin interlayer because it can only occur within 1~2 nm [Appl. Phys. Lett. 2007, 91, 023505; Appl. Phys. Lett. 2006, 89, 083509]. Schwartz et al. [Appl. Phys. Lett. 2006, 89, 083509.] have demonstrated efficient WOLEDs based on blue fluorescent and orange phosphorescent materials by introducing a bipolar interlayer of co-evaporated 4’,4’, 4’’-tris(N-carbazolyl)- triphenylamine and 2,2’, 2’’-(1,3,5-benzenetriyl) tris-[1-phenyl-1Hbenzimidazole] to suppress energy transfer between the two emitters. In this study, the mixed interlayer of MCP: TmPyPb can not only adjust charge transport but also acts as exciton block layer due to the high triplet energy.

Comment 2: “In Figure 2, the device C with lowest Voc, why? Low recombination or high electron injection?

Our response: As shown in Fig. 1 in the revised manuscript, the interfacial energy barrier for electrons injection from TmPyPb (LUMO: 2.7 ev) to MCP (LUMO: 2.4 ev) is 0.3 eV, resulting in electrons accumulated at the interface and higher turn-on voltage. As the concentration of FIrPic increasing, more electrons are injected into FIrPic (LUMO: 2.9 ev) without interfacial energy barrier. Meanwhile, the trapped holes by FIrPic could also facilitate electron injection into EML [Adv. Funct. Mater. 2009, 19, 84]. The current density markedly increases with the increasing doping concentration, shown in Fig. 4. However, when FIrPic concentration further increases to 20%, the holes trapped effect will get serious, resulting in higher turn-on voltage than device C. To clarify this point, the discussion has been included in the revised manuscript.

Comment 3: “In Figure 3, compare with device C, I found the Device A and D shown more interesting the results, for example, the roll-off of these devices are much lower than the device C. And the device D got the max current efficiency-luminance at high voltage and luminance, why?

Our response: Though device D has lowest efficiency roll-off, the maximum efficiencies are only 22.7 cd/A and 9.4%. The lowest efficiency of device D is attributed to the relative strong emission intensity from FIrPic. Device C shows maximum efficiencies of 31.0 cd/A and 13.1%. In contrast to devices A, B and C, the maximum efficiencies are 26.7 cd/A, 30.7 cd/A and 22.7 cd/A, respectively. Moreover, device C with CIE coordinates (0.38, 0.54) emits warm white light.

Comment 4: “If the 15% is optimized the carrier balance in device, the roll-off also should be lowest in four device, but on the contrary it is highest. Does it mean higher efficiency with lower roll off in this two phosphorescent emission device?

Our response: Device C with the optimized concentration obtains highest efficiencies and low efficiency roll-off. The higher efficiency is attributed to weak blue emission and lower turn-on voltage. Note that the efficiency roll-off of the device D is lower than that of device C. From the response to comment 2, we know that FIrPic can facilitate electron injection and trap holes. The increasing concentration of FIrPic will balance against the major carriers (the holes). So relative balanced carrier injection into emission zone and broader recombination zone could be obtained, leading to lowest efficiency roll-off in device D. In our future research work, we will carry out more experiments to make more in-depth analysis and investigations on the underlying roll-off processes. Thanks again for the Reviewer’s helpful comments.

Comment 5: “Does the Author have another result and experiment data support the electron injection from blue layer to red& green EML, photo-celiv data?

Our response: It is really true as Reviewer suggested that some basic measurements will help to understand the electron injection mechanism. Considering the time limit of the revised manuscript and the length of the letter into account, it is difficult for us to carry out some measurements. Again, we appreciate for the Reviewer’s understanding. The first half of the manuscript has thoroughly rewritten in the revised manuscript. We hope the revision will meet with approval. Thanks again for the Reviewer’s helpful comments.

Round 2

Reviewer 1 Report

Given that the authors addressed most of my points, I think that now the manuscript can be published in Molecules.